# Incidence and Impact of Acute Kidney Injury in Patients Receiving Extracorporeal Membrane Oxygenation: A Meta-Analysis

**DOI:** 10.3390/jcm8070981

**Published:** 2019-07-05

**Authors:** Charat Thongprayoon, Wisit Cheungpasitporn, Ploypin Lertjitbanjong, Narothama Reddy Aeddula, Tarun Bathini, Kanramon Watthanasuntorn, Narat Srivali, Michael A. Mao, Kianoush Kashani

**Affiliations:** 1Division of Nephrology and Hypertension, Mayo Clinic, Rochester, MN 55905, USA; 2Division of Nephrology, Department of Medicine, University of Mississippi Medical Center, Jackson, MS 39216, USA; 3Department of Internal Medicine, Bassett Medical Center, Cooperstown, NY 13326, USA; 4Division of Nephrology, Department of Medicine, Deaconess Health System, Evansville, IN 47747, USA; 5Department of Internal Medicine, University of Arizona, Tucson, AZ 85721, USA; 6Division of Pulmonary and Critical Care Medicine, St. Agnes Hospital, Baltimore, MD 21229, USA; 7Division of Nephrology and Hypertension, Mayo Clinic, Jacksonville, FL 32224, USA; 8Division of Pulmonary and Critical Care Medicine, Department of Medicine, Mayo Clinic, Rochester, MN 55905, USA

**Keywords:** acute kidney injury, AKI, extracorporeal membrane oxygenation, ECMO, epidemiology, meta-analysis

## Abstract

Background: Although acute kidney injury (AKI) is a frequent complication in patients receiving extracorporeal membrane oxygenation (ECMO), the incidence and impact of AKI on mortality among patients on ECMO remain unclear. We conducted this systematic review to summarize the incidence and impact of AKI on mortality risk among adult patients on ECMO. Methods: A literature search was performed using EMBASE, Ovid MEDLINE, and Cochrane Databases from inception until March 2019 to identify studies assessing the incidence of AKI (using a standard AKI definition), severe AKI requiring renal replacement therapy (RRT), and the impact of AKI among adult patients on ECMO. Effect estimates from the individual studies were obtained and combined utilizing random-effects, generic inverse variance method of DerSimonian-Laird. The protocol for this systematic review is registered with PROSPERO (no. CRD42018103527). Results: 41 cohort studies with a total of 10,282 adult patients receiving ECMO were enrolled. Overall, the pooled estimated incidence of AKI and severe AKI requiring RRT were 62.8% (95%CI: 52.1%–72.4%) and 44.9% (95%CI: 40.8%–49.0%), respectively. Meta-regression showed that the year of study did not significantly affect the incidence of AKI (*p* = 0.67) or AKI requiring RRT (*p* = 0.83). The pooled odds ratio (OR) of hospital mortality among patients receiving ECMO with AKI on RRT was 3.73 (95% CI, 2.87–4.85). When the analysis was limited to studies with confounder-adjusted analysis, increased hospital mortality remained significant among patients receiving ECMO with AKI requiring RRT with pooled OR of 3.32 (95% CI, 2.21–4.99). There was no publication bias as evaluated by the funnel plot and Egger’s regression asymmetry test with *p* = 0.62 and *p* = 0.17 for the incidence of AKI and severe AKI requiring RRT, respectively. Conclusion: Among patients receiving ECMO, the incidence rates of AKI and severe AKI requiring RRT are high, which has not changed over time. Patients who develop AKI requiring RRT while on ECMO carry 3.7-fold higher hospital mortality.

## 1. Introduction

Extracorporeal membrane oxygenation (ECMO), as a mechanical circulatory support system, is utilized as a treatment for cardiovascular or respiratory failure [1,2,3]. There are two main types of ECMO, including venovenous (VV)-ECMO for patients with isolated respiratory failure and venoarterial (VA)-ECMO for combined severe cardiac and respiratory failure [4]. Over the past 40 years, the clinical applications and feasibility of ECMO have expanded in patients with refractory cardiorespiratory failure, and there has been an exponential increase in the number of centers utilizing ECMO globally [3,5,6,7,8,9]. Studies have demonstrated survival benefits of ECMO ranging from 20% to 50% in patients with cardiac arrest, severe adult respiratory distress syndrome (ARDS), and refractory cardiogenic shock [5,10,11,12,13,14,15,16]. 

Despite these benefits, there have been a number of reports to highlight the concomitant occurrence of organ failures and complications including acute kidney injury (AKI), infections, thrombosis, bleeding and coagulopathy, and neurological events [17,18]. The underlying mechanisms for AKI among patients requiring ECMO appear to be complex and include hemodynamic instabilities, inflammatory responses, coagulation-platelet abnormalities, and immune-mediated injury that arise from the primary underlying disease, premorbid conditions and the ECMO circuit [18,19,20,21,22,23,24,25,26,27,28]. Due to previously non-uniform definitions of AKI, the reported incidences of AKI among patients requiring ECMO therapy ranged widely from 8% up to 85% [4,7,15,18,19,20,21,22,23,24,25,26,27,28,29,30,31,32,33,34,35,36,37,38,39,40,41,42,43,44,45,46,47,48,49,50,51,52,53,54,55,56,57,58,59,60,61,62,63,64,65,66,67,68,69,70]. In addition, the incidence and mortality associated with AKI in patients requiring ECMO and their trends remain unclear.

This systematic review was conducted with the aim to summarize the incidence (using standard AKI definitions) and the impact of AKI on mortality risk among adult patients on ECMO.

## 2. Methods

### 2.1. Information Sources and Search Strategy 

The protocol for this systematic review and meta-analysis is registered with International Prospective Register of Systematic Reviews (PROSPERO no. CRD42018103527). A systematic literature review of EMBASE, Ovid MEDLINE, and the Cochrane Database of Systematic Reviews from database inception through March 2019 was conducted to summarize the incidence and impact of AKI on mortality risk among adult patients on ECMO. Two authors (C.T. and W.C.) independently performed a systematic literature search utilizing a search approach that consolidated the search terms “extracorporeal membrane oxygenation” OR “ECMO” AND “acute kidney injury” OR “acute renal failure.” Further details regarding the search strategy utilized for each database are provided in Online Appendix A. No language restriction was implemented. A manual search for conceivably related articles utilizing references of the included studies was additionally performed. This systematic review was performed following the PRISMA (Preferred Reporting Items for Systematic Reviews and Meta-Analysis) statement [71].

### 2.2. Study Selection

Studies were included in this systematic review if they were clinical trials or observational studies that reported the incidence of AKI (using standard AKI definitions including RIFLE (Risk, Injury, Failure, Loss of kidney function, and End-stage kidney disease) [72], AKIN (Acute Kidney Injury Network) [73], and KDIGO (Kidney Disease: Improving Global Outcomes) classifications) [74], severe AKI requiring renal replacement therapy (RRT), and mortality risk of AKI among adult patients (age ≥ 18 years old) on ECMO. Eligible studies needed to provide the data to evaluate the incidence or mortality rate of AKI with 95% confidence intervals (CI). Retrieved articles were independently examined for eligibility by the two authors (C.T. and W.C.). Inconsistencies were discussed and resolved by shared agreement. The size of the study did not limit inclusion.

### 2.3. Data Collection Process 

A structured data collecting form was adopted to gather the following data from individual study including title, name of authors, publication year, year of the study, country where the study was conveyed, type of ECMO, AKI definition, incidence of AKI, incidence of severe AKI requiring RRT, and mortality risk of AKI among patients on ECMO.

### 2.4. Statistical Analysis

We used the Comprehensive Meta-Analysis software version 3.3.070 (Biostat Inc, Englewood, NJ, USA) to conduct the meta-analysis. Adjusted point estimates of included studies were consolidated by the generic inverse variance method of DerSimonian-Laird, which assigned the weight of individual study based on its variance [75]. Due to the probability of between-study variance, we applied a random-effects model to pool outcomes of interest, including the incidence of AKI and mortality risk. Statistical heterogeneity of studies was assessed by the Cochran’s Q test (*p* < 0.05 for a statistical significance) and the *I*^2^ statistic (≤25%: insignificant heterogeneity, 26%–50%: low heterogeneity, 51%–75%: moderate heterogeneity and ≥75%: high heterogeneity) [76]. The presence of publication bias was evaluated by both the funnel plot and the Egger test [77].

## 3. Results

A total of 1,632 potentially eligible articles were identified with our search approach. After excluding 644 articles that were either in-vitro studies, focused on pediatric patient population, animal studies, case reports, correspondences, or review articles, and 831 articles due to being duplicates, 157 articles remained for full-length article review. Seventy-three articles were subsequently excluded as they did not provide data on the incidence of AKI or mortality of AKI, while 33 articles were excluded because they were not clinical trials or observational studies. Ten studies [19,20,21,22,23,24,25,26,27,28] were additionally excluded because they did not use a standard AKI definition or did not report the incidence of severe AKI requiring RRT. Therefore, 41 cohort studies [7,15,29,30,31,32,33,34,35,36,37,38,39,40,41,42,43,44,45,46,47,48,49,50,51,52,53,54,55,56,57,58,59,60,61,62,63,64,65,66,67] with a total of 10,282 adult patients receiving ECMO were enrolled. The systematic review of the literature flowchart is demonstrated in Figure 1. The characteristics of the included studies are shown in Table 1.

### 3.1. Incidence of AKI in Patients Requiring ECMO

Overall, the pooled estimated incidence of AKI and severe AKI requiring RRT while on ECMO were 62.8% (95%CI: 52.1%–72.4%, *I*^2^ = 94%, Figure 2A) and 44.9% (95%CI: 40.8%-49.0%, *I*^2^ = 91%, Figure 2B), respectively. Subgroup analyses were performed according to AKI definitions. The pooled estimated incidence rates of AKI by RIFLE, AKIN, and KDIGO criteria were 67.5% (95%CI: 43.9%–84.6%, *I*^2^ = 85%), 57.8% (95%CI: 44.6%–70.0%, *I*^2^ = 86%), and 68.2% (95%CI: 43.8%–85.55%, *I*^2^ = 98%), respectively.

Subgroup analysis based on the type of ECMO was also performed. Pooled estimated incidence of AKI and severe AKI requiring RRT while on venoarterial (VA)-ECMO were 60.8% (95%CI: 32.9%–83.1%, *I*^2^ = 96%) and 49.5% (95%CI: 39.6%–59.4%, *I*^2^ = 90%), respectively. Pooled estimated incidence of AKI and severe AKI requiring RRT while on venovenous (VV)-ECMO were 45.7% (95%CI: 33.2%–58.8%, *I*^2^ = 47%) and 37.0% (95%CI: 14.8%–66.5%, *I*^2^ = 95%), respectively. Meta-regression showed that year of the study did not significantly affect the incidence of AKI (*p* = 0.67) or AKI requiring RRT (*p* = 0.83), as shown in Figure 3.

### 3.2. AKI associated Mortality in Patients Requiring ECMO

Mortality rate and mortality risk associated with AKI in patients requiring ECMO are demonstrated in Table 1 and Table 2, respectively. The pooled estimated hospital and/or 90-day mortality rates of patients with AKI and severe AKI requiring RRT while on ECMO were 62.0% (95%CI: 54.7%–68.8%, *I*^2^ = 73%, Figure 4A) and 68.4% (95%CI: 62.6%–73.6%, *I*^2^ = 87%, Figure 4B), respectively.

The pooled OR of hospital mortality among patients receiving ECMO with AKI on RRT was 3.73 (95% CI, 2.87–4.85, *I*^2^ = 62%, Figure 5A). When the analysis was limited to studies with confounder-adjusted analysis, the increased hospital mortality remained significant among patients receiving ECMO with AKI requiring RRT with pooled OR of 3.32 (95% CI, 2.21–4.99, *I*^2^ = 82%, Figure 5B).

Meta-regression showed that year of the study did not significantly affect hospital mortality among patients receiving ECMO with AKI requiring RRT (*p* = 0.86), as shown in Figure 6.

### 3.3. Evaluation for Publication Bias

Funnel plots (Figure 7) and Egger’s regression asymmetry tests were utilized to assess for publication bias in our meta-analyses evaluating the incidence of AKI and severe AKI requiring RRT while on ECMO. There was no publication bias as determined by the funnel plot and Egger’s regression asymmetry test with *p* = 0.62 and *p* = 0.17 for the incidence of AKI and severe AKI requiring RRT, respectively.

## 4. Discussion

The findings of our meta-analysis demonstrate that patients who required ECMO had incidence rates of AKI (using standard AKI definitions) and severe AKI requiring RRT of 62.8% and 44.9%, respectively. Moreover, patients with AKI and severe AKI requiring RRT had high associated mortality rates of 62.0% and 68.4%, respectively.

Although the mechanisms underlying ECMO associated-AKI remains unclear, it is likely complex and multifactorial, including contributing factors such as primary disease progression, altered hemodynamics, low cardiac output syndrome, exposure to nephrotoxic agents (for management of underlying diseases), new-onset sepsis, high intrathoracic pressures, fluid overload, ischemia-reperfusion injury, release of proinflammatory mediators and oxidative stress, hemolysis and iron-mediated (hemoglobin-induced) renal injury, and hypercoagulable state resulting in renal microembolisms [4,8,68,78,79]. Studies have demonstrated the activation of proinflammatory mediators such as tumor necrosis factor-alpha (TNF-α), interleukins (e.g., IL-1β, IL-6, IL-8) and other cytokine signaling cascades due to the continuous exposure of blood to non-biological and non-endothelialized ECMO interface [68,80,81]. Activation of the inflammatory cascades can result in hyperdynamic vasodilated hypotensive states, leading to AKI [68,78]. 

Following the initiation of ECMO treatment, there are improvements in oxygenation and oxygen consumption as well as hemodynamics [3,5,6,7,8,9]. However, ischemia-reperfusion injury can also occur after the restoration of circulation to previously hypoxic cells and hypoperfused organs, leading to the production of reactive oxygen species (ROS) and oxidative stress-mediated injury [68,78]. In addition, ECMO-associated complications or adverse effects such as hemolysis, hemorrhage or thrombosis also can play important roles in the development of AKI [29,68,82,83,84]. Despite the advance of a new miniaturized ECMO system, hemolysis due to shear stress from the ECMO circuit has been reported among ECMO patients with incidences between 5% and 18% [17,85,86,87]. This can contribute to heme pigment-induced AKI [83,84]. Although improvements in the ECMO technology have led to less thrombus development in its circuit with an improved capacity of the circuit to remove large emboli [68,82], smaller thrombi can still develop and result in renal microembolism [68,82], particularly with VA-ECMO [82]. 

The type of ECMO may also differently affect AKI risk. Our study demonstrated a higher incidence of AKI among patients requiring VA-ECMO (60.8%) than those requiring VV-ECMO (45.7%). While VV-ECMO is typically utilized for patients with isolated respiratory failure, VA-ECMO is used for combined severe cardiac and respiratory failure [4]. In VA-ECMO, there is a mixture of pulsatile arterial flow from the native heart and non-pulsatile arterial flow from the ECMO pump. Conversely, VV-ECMO maintains pulsatile cardiac output, and alterations in renal perfusion may conceivably be smaller [4]. Recent studies have shown that pulsatile flow may provide beneficial effects over non-pulsatile flow, especially protective effects on microcirculation and renal perfusion [88,89,90]. The differences in patient population and pulsatility between the two types of ECMO are likely explanations underlying the higher AKI incidence among patients requiring VA-ECMO.

As there is no treatment available for AKI, management of AKI is limited to appropriate secondary preventive measures and supportive strategies [91,92,93,94,95,96]. RRT in the form of continuous renal replacement therapy (CRRT) is often required among patients requiring ECMO with severe AKI [42,55,97]. Our study demonstrated no significant correlation between the year of study and the incidence of AKI and/or severe AKI requiring RRT despite considerable changes in technology and practice of ECMO among adult patients. Furthermore, we showed a 3.7-fold increased risk of hospital mortality among ECMO patients with severe AKI requiring RRT. Thus, prevention and early identification of AKI among patients at-risk of ECMO-associated AKI could potentially play a crucial role in improved survival. Studies have shown several important AKI risk factors among patients requiring ECMO including older age, elevated lactate levels before ECMO initiation, high dose of inotropic drugs, severely reduced left ventricular ejection fraction, cirrhosis, postcardiotomy shock as an indication for ECMO, and finally ECMO pump speed and its duration [56,64,67]. Lee et al. recently observed a lower AKI association with a higher ECMO pump speed [56]. Although the underlying pathophysiology remains unclear, excessive ECMO pump speed has been shown to induce hemolysis and complement activation in vitro and animal model [98,99]. In pediatric patients receiving ECMO, Lou et al. also demonstrated higher pump speeds are associated with hemolysis and a number of other adverse clinical outcomes [100]. To prevent hemolysis-mediated kidney injury, it is suggested to limit pump revolutions/min (RPM) to safe levels (i.e., 3000 to 3500 RPM) in order to avoid excessive negative pressures generated within the pump [101]. Future prospective studies are required to assess the effects of ECMO pump speed on AKI risk in ECMO patients. In addition, future studies creating risk prediction models for ECMO-associated AKI are needed to assist with the prevention of AKI in a timely manner, which could potentially lead to an improvement in patient survival.

Our study has several limitations. Firstly, there are statistical heterogeneities in our meta-analysis. Potential sources for heterogeneities were the variations in patient characteristics among the included studies. However, we performed subgroup analysis to assess the AKI incidence based on types of ECMO and a separate meta-analysis that only included studies with confounder-adjusted analysis for mortality risk. Another limitation was that AKI diagnosis was mainly based on serum creatinine [102,103,104] while the data on urine output and novel biomarkers for AKI [105,106,107,108] were limited. Lastly, this systematic review is primarily based on observational studies, as the data from clinical trials or population-based studies were limited. Therefore, it can at best, demonstrate an association but not a causal relationship.

## 5. Conclusions

In conclusion, there is an overall high incidence of AKI and severe AKI requiring RRT in ECMO patients of 62.8% and 44.9%, respectively. The incidence of ECMO-associated AKI has not changed over time. AKI requiring RRT while on ECMO is associated with 3.7-fold increased risk of hospital mortality. Future studies should focus on strategies for prediction, detection, and prevention of AKI among patients who receive ECMO.

## Figures and Tables

**Figure 1 jcm-08-00981-f001:**
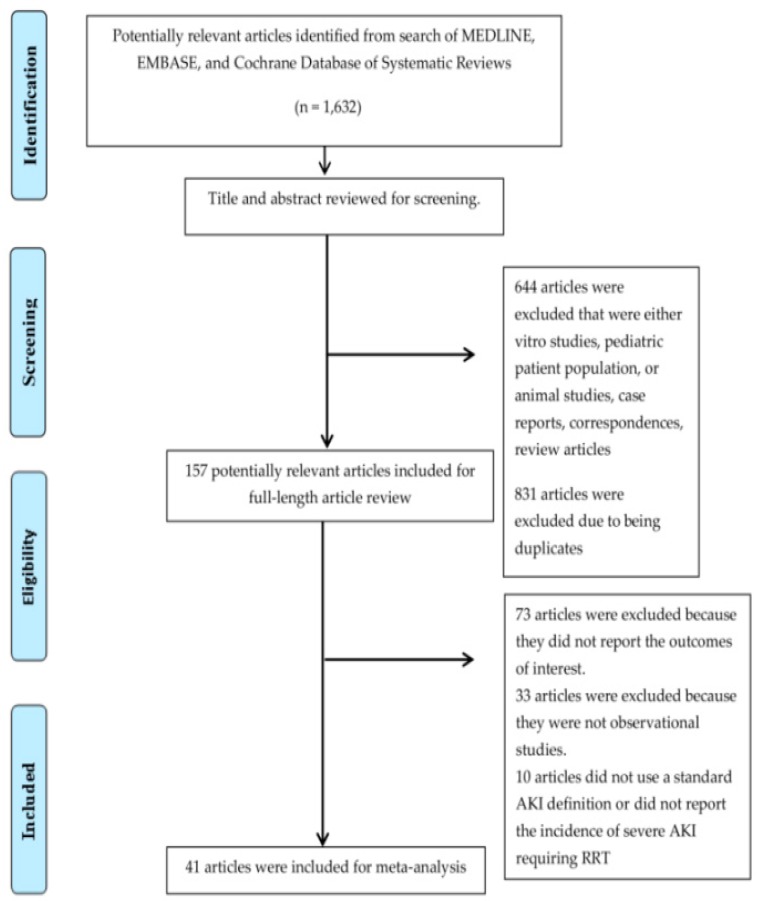
The flowchart for the systematic review.

**Figure 2 jcm-08-00981-f002:**
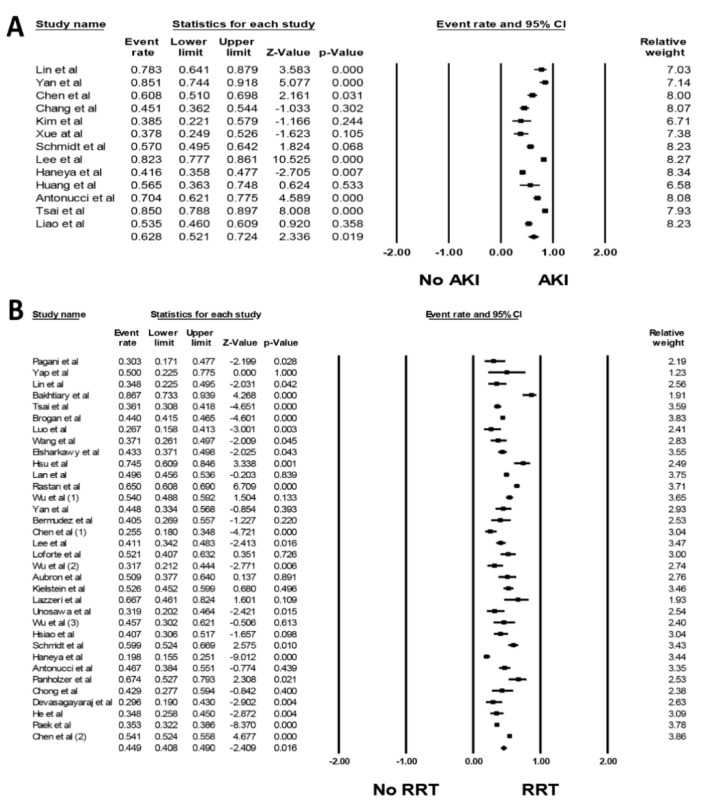
Forest plots of the included studies assessing (**A**) incidence rates of AKI while on ECMO and (**B**) incidence rate of severe AKI requiring RRT while on ECMO. A diamond data marker depicts the overall rate from each included study (square data marker) and 95%CI.

**Figure 3 jcm-08-00981-f003:**
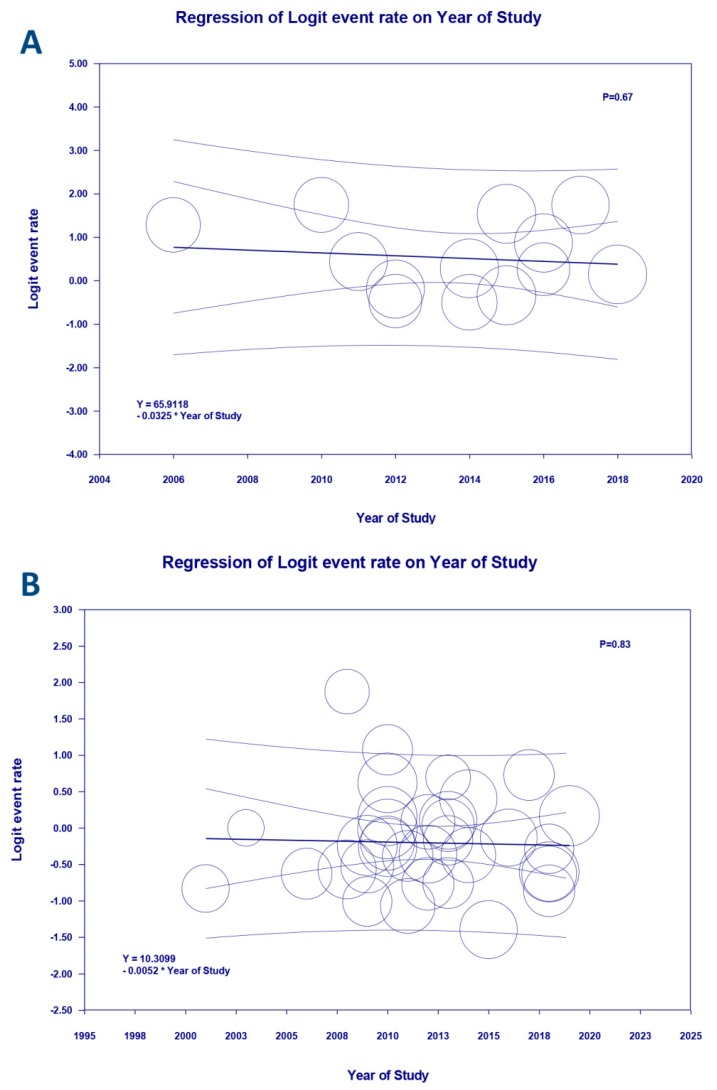
Meta-regression analyses showed that year of the study did not significantly affect (**A**) the incidence of AKI (*p* = 0.67) or (**B**) AKI requiring RRT (*p* = 0.83). The solid black line depicts the weighted regression line based on variance-weighted least squares. The inner and outer lines represent the 95%CI and prediction interval encompassing the regression line. The circles indicate log event rates in individual study.

**Figure 4 jcm-08-00981-f004:**
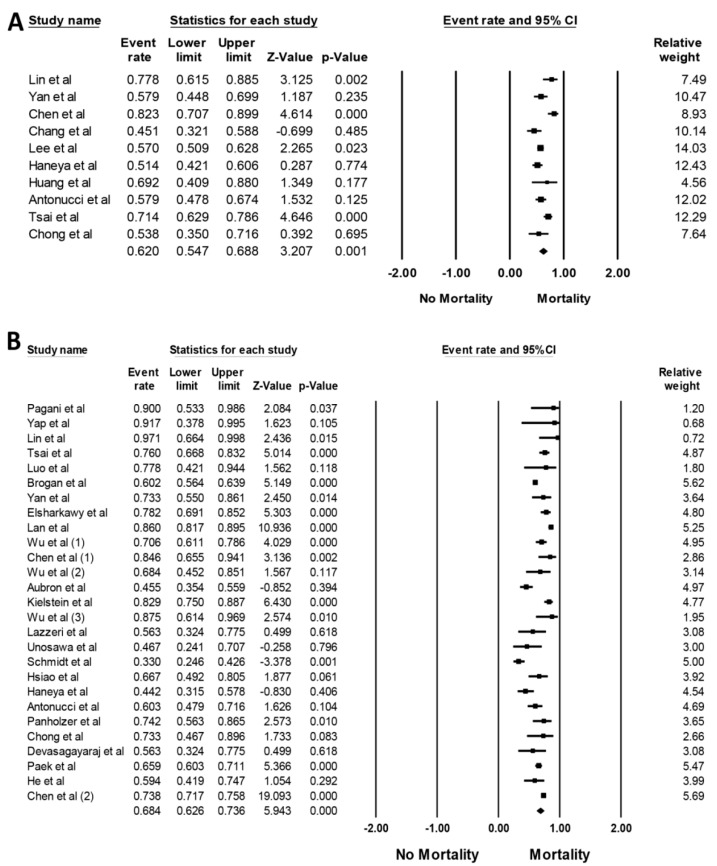
Forest plots of the included studies assessing (**A**) mortality rate of patients with AKI while on ECMO and (**B**) mortality rate of patients with severe AKI requiring RRT while on ECMO. A diamond data label serves as the overall rate from each study (square data marker) and 95%CI.

**Figure 5 jcm-08-00981-f005:**
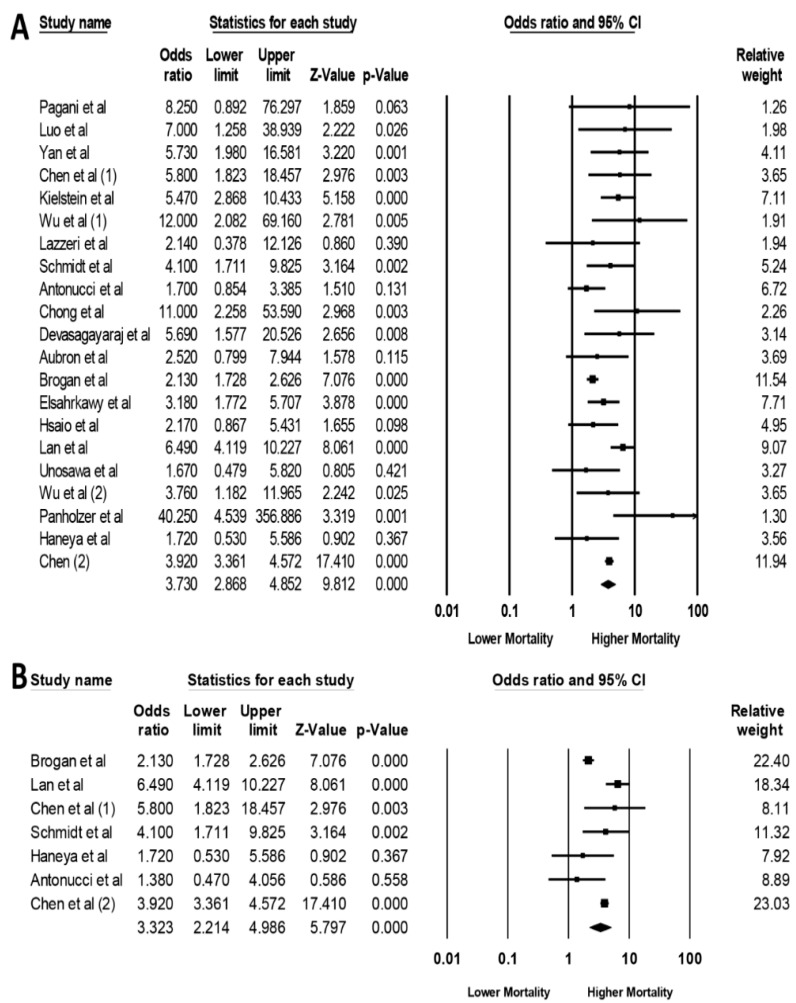
Forest plots of the included studies assessing (**A**) hospital mortality among patients receiving ECMO with AKI on RRT and (**B**) hospital mortality among patients receiving ECMO with AKI on RRT limited to studies with confounder-adjusted analysis. A diamond data label serves as the overall rate from each included study (square data marker) and 95%CI.

**Figure 6 jcm-08-00981-f006:**
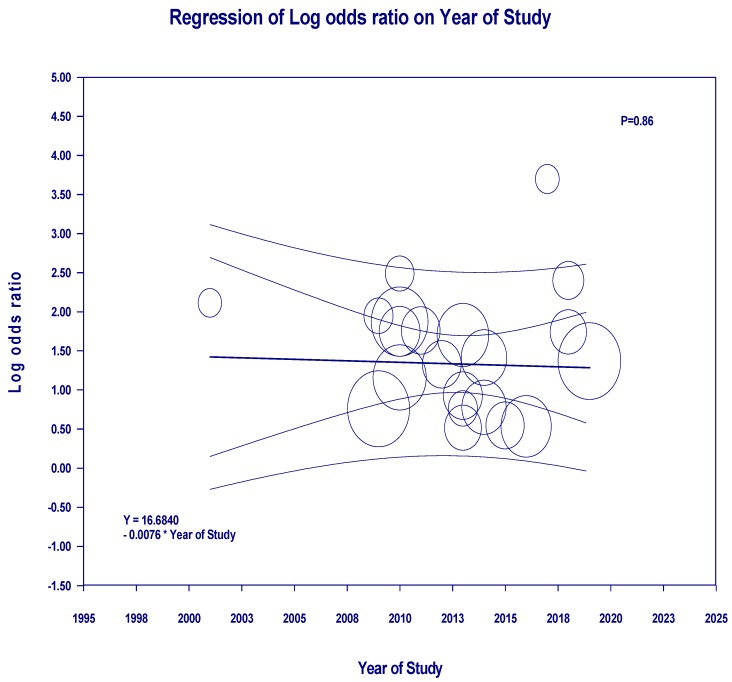
Meta-regression analyses showed that year of the study did not significantly affect hospital mortality among patients receiving ECMO with AKI requiring RRT (*p* = 0.86). The solid black line depicts the weighted regression line based on variance-weighted least squares. The inner and outer lines represent the 95%CI and prediction interval encompassing the regression line. The circles indicate log event rates in an individual study.

**Figure 7 jcm-08-00981-f007:**
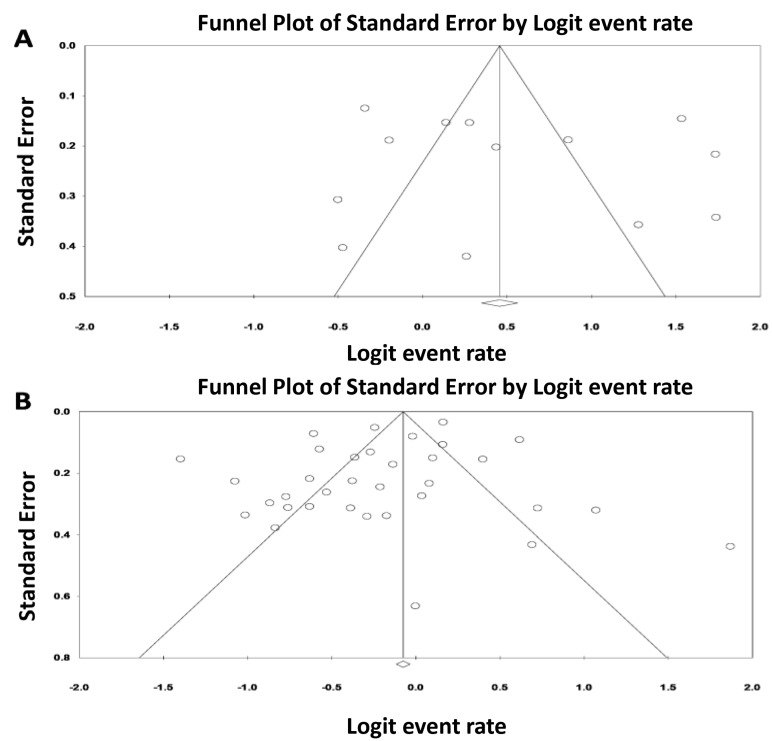
Funnel plot demonstrated no publication bias in analyses evaluating (**A**) incidence of AKI in patients requiring ECMO and (**B**) severe AKI requiring RRT.

**Table 1 jcm-08-00981-t001:** Main characteristic of studies included in this meta-analysis of AKI incidence and mortality among patients requiring ECMO [7,15,29,30,31,32,33,34,35,36,37,38,39,40,41,42,43,44,45,46,47,48,49,50,51,52,53,54,55,56,57,58,59,60,61,62,63,64,65,66,67].

Study	Year	Country	Patients	Number	AKI Definition	AKI Incidence	Mortality
Pagani et al. [15]	2001	USA	ECMO for cardiogenic shock or arrest	33	RRT	RRT 10/33 (30.3%)	Hospital mortality 9/10 (90%)
Yap et al. [29]	2003	Taiwan	ECMO for cardiogenic shock	10	RRT	RRT 5/10 (50%)	Mortality 5/5 (100%)
Lin et al. [30]	2006	Taiwan	ECMO	46	AKI; RIFLE criteria	AKI 36/46 (78.3%) CRRT 16/46 (34.8%)	AKI: Hospital mortality 28/36 (78%) CRRT: Hospital mortality 16/16 (100%)
Tsai et al. [31]	2008	Taiwan	ECMO	288	CRRT	CRRT 104/288 (36.1%)	Hospital mortality 79/104 (76%)
Bakhtiary et al. [32]	2008	Germany	VA-ECMO for refractory cardiogenic shock	45	CRRT	CRRT 39/45 (86.7%)	N/A
Luo et al. [33]	2009	China	VA-ECMO in severe heart failure	45	CRRT	CRRT 12/45 (26.6%)	Hospital mortality 7/9 (78%)
Brogan et al. [34]	2009	USA	ECMO in severe respiratory failure	1473	RRT	RRT 648/1473 (44%)	Hospital mortality RRT 390/648 (60%)
Wang et al. [35]	2009	China	VA ECMO for refractory cardiogenic shock after cardiac surgery	62	CRRT	CRRT 23/62 (37.0%)	N/A
Yan et al. [36]	2010	China	ECMO after cardiac surgery	67	AKI; RIFLE and AKIN criteria	RIFLE AKI 54/67 (80.6%) AKIN AKI 57/67 (85.1%) RRT 30/67 (44.8%)	Hospital mortality RIFLE AKI 32/54 (59%) AKIN AKI 33/57 (58%) RRT 22/30 (73%)
Elsharkawy et al. [37]	2010	USA	VA-ECMO after cardiac surgery	233	RRT	RRT 101/233 (43.3%)	Hospital mortality 79/101 (78%)
Hsu et al. [38]	2010	Taiwan	VA-ECMO for cardiogenic shock after cardiac surgery	51	CRRT	CRRT 38/51 (74.5%)	N/A
Lan et al. [39]	2010	Taiwan	ECMO	607	RRT	RRT 301/607 (49.6%)	Hospital mortality 259/301 (86%)
Rastan et al. [40]	2010	Germany	VA-ECMO for cardiogenic shock after cardiac surgery	517	RRT	RRT 336/517 (65.0%)	N/A
Wu et al. [41]	2010	Taiwan	ECMO	346	RRT	RRT 187/346 (54%)	RRT 72/102 (71%)
Chen et al. [42]	2011	Taiwan	ECMO	102	AKI; AKIN criteria	AKI 62/102 (60.8%) CRRT 26/102 (25.5%)	Hospital mortality AKI 51/62 (82%) CRRT 22/26 (85%)
Bermudez et al. [43]	2011	USA	ECMO for refractory cardiogenic shock; VA (88%)	42	RRT	RRT 17/42 (40.5%)	N/A
Chang et al. [44]	2012	Taiwan	Successfully weaned from ECMO	113	AKI; AKIN criteria at 48 h post-ECMO removal	AKI 51/113 (45.1%)	Hospital mortality AKI 23/51 (45%)
Kim et al. [45]	2012	Korea	ECMO; VA-ECMO (85%), VV-ECMO (15%)	26	AKI; AKIN criteria	AKI 10/26 (38.5%)	N/A
Lee et al. [46]	2012	Korea	ECMO; VA-ECMO (74%), VV-ECMO (26%)	185	CRRT	CRRT 76/185 (41.1%)	N/A
Loforte et al. [47]	2012	Italy	VA-ECMO	73	CRRT	CRRT 38/73 (52.1%)	N/A
Wu et al. [48]	2012	Taiwan	ECMO for non-post cardiotomy cardiogenic shock or cardiac arrest	60	RRT	RRT 19/60 (31.7%)	Hospital mortality 13/19 (68%)
Aubron et al. [49]	2013	Australia	ECMO; VA-ECMO (67%), VV-ECMO (33%)	158	RRT	VA-ECMO RRT 61/105 (58.1%) VV-ECMO RRT 27/53 (50.9%)	Hospital mortality VA-ECMO RRT 27/61 (44%) VV-ECMO RRT 13/27 (48%)
Kielstein et al. [50]	2013	Germany	ECMO; VA-ECMO (45%), VV-ECMO (55%)	200	RRT	RRT 117/200 (58.5%) RRT after ECMO 92/175 (52.6%)	90-day mortality 97/117 (83%)
Wu et al. [51]	2013	Taiwan	ECMO for acute myocardial infarction-induced cardiac arrest	35	RRT	RRT 16/35 (45.7%)	Hospital mortality 14/16 (88%)
Lazzeri et al. [52]	2013	Italy	ECMO for refractory cardiac arrest	25	RRT	RRT 16/24 (66.7%)	Mortality 9/16 (56%)
Unosawa et al. [53]	2013	Japan	VA-ECMO for refractory cardiogenic shock after cardiac surgery	47	RRT	RRT 15/47 (31.9%)	Mortality on ECMO 7/15 (46.7%)
Xue et al. [54]	2014	China	ECMO in lung transplantation	45	AKI; AKIN criteria	AKI 17/45 (37.8%)	N/A
Schmidt et al. [7]	2014	Australia	ECMO for refractory cardiogenic shock or acute respiratory failure	172	AKI; RIFLE criteria	AKI at ECMO day 1 98/172 (57.0%) CRRT during ECMO 103/172 (59.9%)	90-day mortality CRRT 34/103 (33%)
Hsiao et al. [55]	2014	Taiwan	ECMO for ARDS	81	CRRT	CRRT 33/81 (40.7%)	Hospital mortality CRRT 22/33 (67%)
Lee et al. [56]	2015	Korea	ECMO; VA-ECMO (71%), VV-ECMO (29%)	322	AKI; KDIGO criteria	AKI 265/322 (82.3%)	Hospital mortality 151/265 (57%)
Haneya [57]	2015	Germany	VV-ECMO for ARDS	262	AKI; KDIGO criteria	AKI 109/262 (41.6%) RRT during ECMO 52/262 (19.8%)	Mortality AKI 56/109 (51%) RRT during ECMO 23/52 (44%)
Huang et al. [58]	2016	China	ECMO for acute respiratory distress syndrome; VA-ECMO (17%), VV-ECMO (83%)	23	AKI; AKIN criteria	AKI 13/23 (56.5%)	Mortality 9/13 (69%)
Antonucci et al. [59]	2016	Belgium	ECMO; VA-ECMO (59%), VV-ECMO (41%)	135	AKI; AKIN criteria	AKI 95/135 (70.4%) CRRT 63/135 (46.7%)	ICU mortality AKI 55/95 (58%) CRRT 38/63 (60%)
Tsai et al. [60]	2017	Taiwan	ECMO	167	AKI; RIFLE, AKIN and KDIGO on ECMO day 1	RIFLE AKI 126/167 (75.4%) AKIN AKI 141/167 (84.4%) KDIGO AKI 142/167 (85.0%)	Hospital mortality RIFLE AKI 85/126 (67%) AKIN AKI 90/126 (71%) RIFLE AKI 90/126 (71%)
Panholzer et al. [61]	2017	Germany	VV-ECMO for ARDS	46	RRT	RRT 31/46 (67.4%)	Mortality RRT 23/31 (74%)
Chong et al. [62]	2018	Taiwan	VA-ECMO for acute fulminant myocarditis and cardiogenic shock	35	AKI; not specified	AKI 26/35 (74.3%) RRT 15/35 (42.9%)	Hospital mortality AKI 14/26 (54%) RRT 11/15 (73%)
Devasagayaraj et al. [63]	2018	USA	VV-ECMO for ARDS	54	CRRT	CRRT 16/54 (29.6%)	Hospital mortality 9/16 (56%)
Liao et al. [64]	2018	China	ECMO; VA-ECMO (93%), VV-ECMO (7%)	170	AKI; KDIGO criteria	AKI 91/170 (53.5%)	N/A
Paek et al. [65]	2018	Korea	ECMO	538	CRRT	CRRT 296/838 (35.3%)	30-day mortality 195/296 (66%)
He et al. [66]	2018	China	ECMO	92	CRRT	CRRT 32/92 (34.8%)	Hospital mortality 19/32 (59%)
Chen et al. [67]	2019	Taiwan	ECMO	3251	RRT	RRT 1759/3251 (54.1%)	Hospital mortality 1298/1759 (74%)

Abbreviations: AKI, acute kidney injury; ARDS, acute respiratory distress syndrome; AKIN, Acute Kidney Injury Network; CRRT, continuous renal replacement therapy; ECMO, Extracorporeal membrane oxygenation; ICU, intensive care unit; KDIGO, Kidney Disease Improving Global Outcomes; N/A, not available; RIFLE, Risk, Injury, Failure, Loss of kidney function, and End-stage kidney disease; RRT, Renal replacement therapy; USA, United States of America; VA-ECMO, venoarterial extracorporeal membrane oxygenation; VV-ECMO, venovenous extracorporeal membrane oxygenation.

**Table 2 jcm-08-00981-t002:** Characteristics of studies included in this meta-analysis of AKI associated mortality risk among patients requiring ECMO.

Study.	Year	Number	Outcomes	Confounder Adjustment
Pagani et al. [15]	2001	33	Hospital mortality 8.25 (0.89–76.12)	None
Lin et al. [30]	2006	46	Hospital mortality AKI: 14.0 (2.46–79.55) CRRT: 16/16 vs. 14/30	None
Luo et al. [33]	2009	45	Hospital mortality CRRT: 7.0 (1.26–38.99)	None
Brogan et al. [34]	2009	1473	Hospital mortality Renal insufficiency/failure: 2.13 (1.69–2.72) RRT: 2.13 (1.73–2.63)	Age, duration of mechanical ventilation, weight, pre-ECMO pH, race, diagnosis, ECMO mode, post-ECMO complication
Elsharkawy et al. [37]	2010	233	Hospital mortality RRT: 3.18 (1.77–5.70)	None
Yan et al. [36]	2010	67	Hospital mortality RIFLE AKI: 8.0 (1.61–39.68) AKIN AKI: 12.38 (1.47–104.33) CRRT: 5.73 (1.98–16.58)	None
Lan et al. [39]	2010	607	Hospital mortality RRT: 6.49 (4.12–10.23)	Age, stroke, pre-ECMO infection, hypoglycemia, alkalosis
Chen et al. [67]	2011	102	Hospital mortality AKI: 4.32 (1.65–11.30) CRRT: 5.80 (1.82–18.43)	Age, GCS
Chang et al. [44]	2012	113	Hospital mortality AKI: 2.1 (1.48–3.00)	None
Wu et al. [48]	2012	60	Hospital mortality RRT: 3.76 (1.18–11.95)	None
Kielstein et al. [50]	2013	200	90-day mortality RRT: 5.47(2.87–10.44)	None
Aubron et al. [49]	2013	158	VA ECMO RRT: 2.12 (0.92–4.88) VV ECMO RRT: 2.52 (0.80–7.95)	None
Wu et al. [51]	2013	35	Hospital mortality RRT: 12 (2.08–69.09)	None
Slottosch et al. [23]	2013	77	30-day mortality Renal failure: 2.20 (0.78–6.12)	None
Unosawa et al. [53]	2013	47	Mortality during ECMO RRT: 1.67 (0.48–5.83)	None
Lazzeri et al. [52]	2013	25	Mortality RRT: 2.14 (0.38–12.20)	None
Hsiao et al. [55]	2014	81	Hospital mortality CRRT: 2.17 (0.87–5.45)	None
Schmidt et al. [7]	2014	172	Hospital mortality CRRT at ECMO day 1–3: 4.1 (1.71–9.82) 90-day mortality CRRT at ECMO day 1–3: 3.17 (1.32–7.61)	APACHE, fluid balance, major bleeding, propensity score
Lee et al. [56]	2015	322	Hospital mortality AKI: 3.71 (1.96–7.02)	None
Haneya et al. [57]	2015	262	Mortality AKI: 2.18 (1.31–3.61) RRT during ECMO: 1.72 (0.53–5.59)	Age, SOFA score, minute volume, pH, lactate, RRT prior to ECMO, RBC, and FFP transfusion
Huang et al. [58]	2016	23	Mortality AKI: 20.25 (1.88–218.39)	None
Lyu et al. [27]	2016	84	Mortality ARF: 23.90 (7.00–81.60)	None
Antonucci et al. [59]	2016	135	ICU mortality AKI: 1.86 (0.88–3.93) CRRT: 1.70 (0.85–3.37)	None
Tsai et al. [60]	2017	167	Hospital mortality RIFLE AKI: 8.55 (3.63–20.16) AKIN AKI: 13.53 (3.87–47.28) KDIGO AKI: 12.69 (3.62–44.46)	None
Panholzer et al. [61]	2017	46	Mortality RRT: 40.25 (4.54–356.93)	None
Martucci et al. [28]	2017	82	Mortality on ECMO AKI stage 3: 4.55 (1.37–15.17)	None
Chong et al. [62]	2018	35	Hospital mortality AKI: 9.33 (1.02–85.70) RRT: 11.0 (2.26–53.64)	None
Devasagayaraj et al. [63]	2018	54	Hospital mortality RRT: 5.69 (1.58–20.56)	None
Chen et al. [67]	2019	3,251	Hospital mortality RRT: 3.92 (3.36–4.57)	Age, sex, ECMO indication, comorbid conditions, hospital level, study year

Abbreviations: AKI, acute kidney injury; ARDS, acute respiratory distress syndrome; AKIN, Acute Kidney Injury Network; APACHE, Acute Physiology and Chronic Health Evaluation; CRRT, continuous renal replacement therapy; ECMO, Extracorporeal membrane oxygenation; FFP, fresh frozen plasma; GCS, Glasgow Coma Scale/Score; ICU, intensive care unit; KDIGO, Kidney Disease Improving Global Outcomes; N/A, not available; RIFLE, Risk, Injury, Failure, Loss of kidney function, and End-stage kidney disease; RBC, red blood cells; RRT, Renal replacement therapy; pH, potential hydrogen; SOFA, Sequential Organ Failure Assessment; VA-ECMO, venoarterial extracorporeal membrane oxygenation; VV-ECMO, venovenous extracorporeal membrane oxygenation.

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
