# Peer review of "Incidence and Impact of Acute Kidney Injury in Patients Receiving Extracorporeal Membrane Oxygenation: A Meta-Analysis"

_jcm, 2019, doi:10.3390/jcm8070981_

Round 1
Reviewer 1 Report
It is an interesting review paper proposed to address a critical clinical problem. In general, a meta review paper is featured with answering a controversial issue which somehow to provide solid evidence to assist experts to make decisions. A major concern of this ms is, since it is clear ECMO induces AKI in the clinic, what is the purpose to perform this meta-review? As well, unpublished data should be included either.
Author Response
Reviewer #1
It is an interesting review paper proposed to address a critical clinical problem. In general, a meta review paper is featured with answering a controversial issue which somehow to provide solid evidence to assist experts to make decisions.
Response: We thank you for reviewing our manuscript and for your critical evaluation. We really appreciated your input and found your suggestions very helpful.
Comment#1 A major concern of this ms is, since it is clear ECMO induces AKI in the clinic, what is the purpose to perform this meta-review?
Response: We greatly appreciated the reviewer’s important input on this important point. Although it is known that AKI commonly occurs in patients requiring ECMO, the incidence and impact of AKI on mortality among patients on ECMO were unclear. Thus, we conducted this systematic review to summarize the incidence and impact of AKI (using a standard AKI definition) on mortality risk among adult patients on ECMO. Ultimately, we are hoping that epidemiological data and mortality risk of ECMO associated AKI from this study will help future researchers to develop strategies/interventions to reduce or improve AKI in this patient population. In addition, we respected the reviewer and we additionally look into the incidence based on type of ECMO (VA vs VV ECMO). The following text in bold has been added in the results of manuscript.
“Subgroup analysis based on the type of ECMO was also performed. Pooled estimated incidence of AKI and severe AKI requiring RRT while on venoarterial (VA)-ECMO were 60.8% (95%CI: 32.9%-83.1%, I2 = 96%) and 49.5% (95%CI: 39.6%-59.4%, I2 = 90%), respectively. Pooled estimated incidence of AKI and severe AKI requiring RRT while on venovenous (VV)-ECMO were 45.7% (95%CI: 33.2%-58.8%, I2 = 47%) and 37.0% (95%CI: 14.8%-66.5%, I2 = 95%), respectively.”
Comment#2 As well, unpublished data should be included either.
Response: We appreciated the reviewer’s helpful comments. One of search databases is also EMBASE, which also included abstracts and unpublished data. However, those abstracts were duplicated with the published data in full text manuscripts. We respected the reviewer’s important point and thus, we performed assessment of publication bias by the funnel plot and Egger's regression asymmetry test. There was no publication bias in our study as evaluated by the funnel plot and Egger's regression asymmetry test with P= 0.62 and P= 0.17 for the incidence of AKI and severe AKI requiring RRT, respectively.
All authors thank the Editors and reviewers for their valuable suggestions. The manuscript has been improved considerably by the suggested revisions!

Reviewer 2 Report
In the Discussion, the author described that "Another limitation was that AKI diagnosis was mainly based on serum creatinine while urine output was commonly not used." The authors should add novel biomarkers for AKI such as urinary Kim-1 (Am J Physiol Renal Physiol. 2006 Feb;290(2):F517-29.), NGAL (J Am Soc Nephrol. 2003 Oct;14(10):2534-43.), and vanin-1 (Pharmacol Exp Ther. 2012 Jun;341(3):656-62.).
Author Response
Reviewer #2
We thank you for reviewing our manuscript and for your critical evaluation. We really appreciated your input and found your suggestions very helpful.
Comment#1
In the Discussion, the author described that "Another limitation was that AKI diagnosis was mainly based on serum creatinine while urine output was commonly not used." The authors should add novel biomarkers for AKI such as urinary Kim-1 (Am J Physiol Renal Physiol. 2006 Feb;290(2):F517-29.), NGAL (J Am Soc Nephrol. 2003 Oct;14(10):2534-43.), and vanin-1 (Pharmacol Exp Ther. 2012 Jun;341(3):656-62.).
Response: We appreciated the reviewer’s important input. We agree with your comment that we should add the lack of novel biomarkers for AKI data in our limitation. Thus, we have added this important point in the limitations and included the reviewers’ reference suggestions as our references (Am J Physiol Renal Physiol. 2006 Feb;290(2):F517-29.), NGAL (J Am Soc Nephrol. 2003 Oct;14(10):2534-43.), and vanin-1 (Pharmacol Exp Ther. 2012 Jun;341(3):656-62.) as well. The following text in bold has been added in the limitation of our revised manuscript.
“Our study has several limitations. Firstly, there are statistical heterogeneities in our meta-analysis. Potential sources for heterogeneities were the variations in patient characteristics among the included studies. However, we performed subgroup analysis to assess the AKI incidence based on types of ECMO and a separate meta-analysis that only included studies with confounder-adjusted analysis for mortality risk. Another limitation was that AKI diagnosis was mainly based on serum creatinine while the data on urine output and novel biomarkers for AKI [105-108] were limited. Lastly, this systematic review is primarily based on observational studies, as the data from clinical trials or population-based studies were limited. Therefore, it can at best, demonstrate an association but not a causal relationship.”
All authors thank the Editors and reviewers for their valuable suggestions. The manuscript has been improved considerably by the suggested revisions!

Round 2
Reviewer 1 Report
The authors addressed most of the concerns.